# Early Administration of Convalescent Plasma Improves Survival in Patients with Hematological Malignancies and COVID-19

**DOI:** 10.3390/v13030436

**Published:** 2021-03-08

**Authors:** Monika Maria Biernat, Anna Kolasińska, Jacek Kwiatkowski, Donata Urbaniak-Kujda, Paweł Biernat, Justyna Janocha-Litwin, Małgorzata Szymczyk-Nużka, Dawid Bursy, Elżbieta Kalicińska, Krzysztof Simon, Grzegorz Mazur, Tomasz Wróbel

**Affiliations:** 1Department and Clinic of Haematology, Blood Neoplasms, and Bone Marrow Transplantation, Wroclaw Medical University, Pasteura Street 4, 50-367 Wroclaw, Poland; ana.jerczynska@gmail.com (A.K.); j.k.kwiatkowski@wp.pl (J.K.); donata.urbaniak-kujda@umed.wroc.pl (D.U.-K.); elzbieta.kalicinska@umed.wroc.pl (E.K.); tomasz.wrobel@umed.wroc.pl (T.W.); 2Department of Drugs Form Technology, Wroclaw Medical University, Borowska Street 211A, 50-556 Wroclaw, Poland; pawel.biernat@umed.wroc.pl (P.B.); dawid.bursy@umed.wroc.pl (D.B.); 3Department of Infectious Diseases and Hepatology, Wroclaw Medical University, Koszarowa Street 5, 51-149 Wroclaw, Poland; justyna.janocha-litwin@umed.wroc.pl (J.J.-L.); krzysztof.simon@umed.wroc.pl (K.S.); 4Regional Centre of Transfusion Medicine and Blood Bank (RCTMBB-W), Czerwonego Krzyża Street 5/9, 50-345 Wroclaw, Poland; malgorzata.szymczyknuzka@rckik.wroclaw.pl; 5Department and Clinic of Internal and Occupational Diseases, Hypertension and Clinical Oncology, Wroclaw Medical University, Borowska Street 213, 50-556 Wroclaw, Poland; grzegorz.mazur@umed.wroc.pl

**Keywords:** COVID-19, convalescent plasma, hematological malignancies

## Abstract

The use of convalescent plasma in the treatment of COVID-19 may lead to a milder course of infection and has been associated with improved outcomes. Determining optimal treatments in high risk populations is crucial, as is the case in those with hematological malignancies. We analyzed a cohort of 23 patients with hematological malignancies and COVID-19 who had received plasma 48–72 h after the diagnosis of infection and compared it with a historical group of 22 patients who received other therapy. Overall survival in those who received convalescent plasma was significantly higher than in the historical group (*p* = 0.03460). The plasma–treated group also showed a significantly milder course of infection (*p* = 0.03807), characterized by less severe symptoms and faster recovery (*p* = 0.00001). In conclusion, we have demonstrated that convalescent plasma is an effective treatment and its early administration leads to clinical improvement, increased viral clearance and longer overall survival in patients with hematological malignancies and COVID-19. To our knowledge, this is the first report to analyze the efficacy of convalescent plasma in a cohort of patients with hematological malignancies.

## 1. Introduction

Patients with hematological malignancies are often in an immunocompromised state, and a high mortality due to COVID-19 in this population has recently been demonstrated, both in our group and elsewhere [1,2]. SARS-CoV-2 infection can complicate the course of standard treatment of hematologic malignancies by delaying subsequent cycles of chemotherapy, worsening the already severe prognosis in these patients [3,4]. Some reports have shown that the use of convalescent plasma in patients with COVID-19 may lead to a milder course of infection and improved outcomes [5,6]. Infusion of convalescent plasma from recovered donors has been successfully used in the past to treat the MERS and SARS coronaviruses as well as other viruses such as measles, Ebola, and influenza A H1N1 [7,8]. However, reports on the use of convalescent plasma in the therapy of patients with COVID-19 have shown conflicting results [5,9,10]. In patients with hematological malignancies, reports on this subject are limited to scarce case reports [11,12]. Given the lack of available effective treatment, we started to treat our patients with convalescent plasma in the second wave of the epidemic, and here we analyze the effectiveness of this treatment in patients with COVID-19, the clinical course and overall survival (OS) compared to a historical group of patients with COVID-19 who did not receive it. To our knowledge, this is the first report to analyze the efficacy and safety of convalescent plasma in those with hematological malignancies.

## 2. Materials and Methods

In this study, a cohort of 45 patients with COVID-19 were analyzed. The group consisted of 23 patients treated in our department between 1 September and 30 November 2020 and a historical group of 22 patients with COVID-19 treated in the first wave of the outbreak between 1 March and 31 May 2020. Infection in all patients was confirmed by RT-PCR test from nasopharyngeal swab, repeated every seven days until a negative result was obtained. In the treatment group, at least one plasma dose of 200–250 mL was administered 48–72 h after the diagnosis of infection. Convalescent plasma transfusion was administered at approximately 10 mL for the first 10 min, which was then increased to approximately 200 mL per 30 min. The patients were carefully monitored for 24 h to assess early post-transfusion reactions as well as late reactions throughout their hospitalization. Plasma donor qualification and collection followed a standard procedure according to European guidelines [13]. Additional analyses on plasma from COVID-19-recovered patients included testing for anti-S IgG SARS-CoV-2 antibody titers using ELISA assay (Elecsys^®^ Anti-SARS-CoV-2, ROCHE). For our purposes, only plasma units with an S-RBD-specific IgG titer greater than 1:1000 in serum were eligible.

All 45 patients were prospectively examined at the following time points: day of infection confirmation at 0, 7, 14, 21, and 28 days after disease onset. Both groups of patients were followed from the onset of infection until recovery or death. Clinical characteristics, laboratory data, and outcome in both groups were then analyzed. All patients had given their written consent to participate and to have their data published. The study was approved by the Wroclaw Medical University Ethics Committee (Consent no. 263/2020).

The analyzed variables were both nominal (including dichotomous) and interval in nature. First, correlations between analyzed factors were assessed using PCA analysis. The PCA model was estimated using the NIPALS iterative algorithm, and the convergence criterion was set at the level of 0.00001. Guided by the results of PCA principal component analysis, the primary grouping variable (Plasma YES/NO) was determined. Univariate logistic regression with a free expression ≠ 0 was used to determine its correlation with the other key variables of the study. The parameters of the logit function were estimated by the least squares method using the Quasi-Newton algorithm. The criterion of convergence for the performed logistic regression was set at the level of 0.00001. Odds ratios (OR) were calculated for each comparison along with their 95% significance interval. The statistical significance of the logistic models was evaluated using the chi-square test for degrees of freedom df = 1 and an assumed significance level α = 0.05. A nonparametric survival analysis based on the nonparametric Cox proportional hazards model was also performed. In all tests, a significance level of α = 0.05 was assumed; the proportionality assumption of the Cox model was evaluated using the graphical method. Statistical analyses were performed using the computer program STATISTICA PL^®^ version 13.

## 3. Results and Discussion

In this study, 23 patients with COVID-19 and hematologic malignancies that received plasma (treatment group) were compared to a historical group of 22 patients (control group) who received other therapy. These groups were similar in age, number and diagnosis of the underlying disease (Table 1).

In the treatment group, 21 patients received one plasma transfusion and supportive care, whereas two patients received two doses of convalescent plasma; remdesivir and other drugs used to treat COVID-19 were not available in our department at that time. In the control group, patients mainly received hydroxychloroquine, remdesivir, lopinavir/ritonavir, tocilizumab and supportive care. A similar number of patients in both groups received dexamethasone. Overall survival in the control group was significantly lower than in the treatment group—this was particularly observed after day 14 (4 deaths vs. 0) and after day 21 (*p* = 0.03460) (Figure 1).

In addition, the group of patients who received convalescent plasma showed a statistically significant milder course of infection (*p* = 0.03807) (Figure 2), with less severe and faster resolution of symptoms such as fever (*p* = 0.00665), shortness of breath (*p* = 0.03008) and cough (*p* = 0.00763).

Statistically significant differences in COVID-19 severity were observed between groups depending on the therapy administered (convalescent plasma vs. other therapies: hydroxychloroquine, tocilizumab, remdesivir, lopinavir/ritonavir, only supportive care). 

Moreover, patients treated with convalescent plasma had a faster clearance of the virus by day 14. Pulmonary infiltrates resolved after day 14, significantly faster than in the control group (*p* = 0.02480) and patients required oxygen therapy for a shorter time, on average 14 days (*p* = 0.02355), and more patients in the treatment group had recovered by day 14 (*p* = 0.00001). Convalescent plasma administration was well tolerated and no adverse events were reported in any case. Our results are consistent with the observations of other authors that the administration of convalescent plasma in patients with COVID-19 pneumonia, including those with a severe clinical course of this infection, showed improvement in their clinical status and resulted in faster clearance of the virus [5,14,15]. A recent report on the use of convalescent plasma in the early phase of infection in elderly patients with COVID-19 also confirmed its efficacy, and it should be emphasized that our group contained mainly elderly patients [15]. In contrast, the study by Li et al. in patients with severe COVID-19 did not demonstrate a benefit with the addition of convalescent plasma to standard treatment [9]. In our study, the highest number of deaths was observed in patients with acute leukemia and chronic lymphocytic leukemia (CLL). Thus, it seems that these patients would stand to benefit most from this treatment. However, it should be noted that our patients with CLL died despite the early use of convalescent plasma. Avanzato et al. demonstrated the presence of prolonged shedding of SARS-CoV-2 virus on days 85 and 105 in a CLL patient, despite two transfusions of convalescent plasma and standard therapy, suggesting that this therapy was not successful in obtaining rapid viral clearance in this individual [16].

Our study has several limitations. First, the study was conducted in a small cohort of patients, with subgroups treated in the first and second wave of the pandemic. International multicenter randomized trials in patients with hematologic malignancies are needed. Since there is still no effective treatment for COVID-19, the administration of plasma from recovered patients may help protect patients with hematological malignancies from severe complications and death, as shown in our study. Secondly, our study concerns a heterogeneous patient population with various different hematologic malignancies at different stages of anticancer treatment; further studies are needed to optimize the best time and dosage of convalescent plasma alone or in combination with other antiviral drugs in patients with specific malignancies. Moreover, the control group was treated for COVID-19 with multiple other drugs, including antiviral medications, antibiotics and corticosteroids. It is not entirely clear whether administering convalescent plasma with other drugs would change the outcome and what the optimal length of treatment would be. Additionally, most patients received only one transfusion of plasma. Whether multiple transfusions would be more effective needs to be determined. Nevertheless, in these patients, there is a need for rapid diagnosis of the SARS-CoV-2 infection and rapid implementation of treatment, and the early administration of convalescent plasma can make a significant difference to the fate of these patients.

## 4. Conclusions

In conclusion, we have demonstrated that convalescent plasma is an effective treatment and its early administration leads to clinical improvement, increased viral clearance and longer overall survival in patients with hematological malignancies with COVID-19.

## Figures and Tables

**Figure 1 viruses-13-00436-f001:**
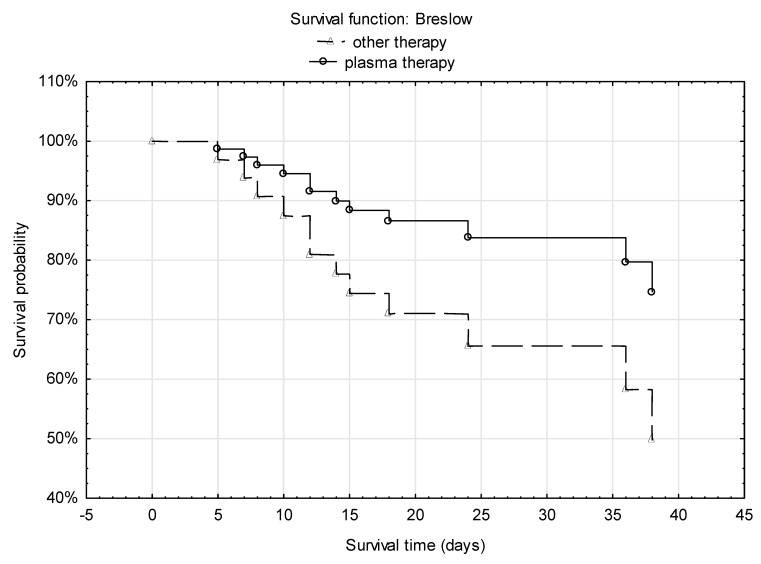
Kaplan–Meier analysis of hematological patients with COVID-19 who received convalescent plasma compared with patients from a historical group treated with other therapy (remdesivir, tocilizumab, hydroxychloroquine, lopinavir/ritonavir). Hazard ratio (HR) and 95% CI are calculated from a Cox model without covariates.

**Figure 2 viruses-13-00436-f002:**
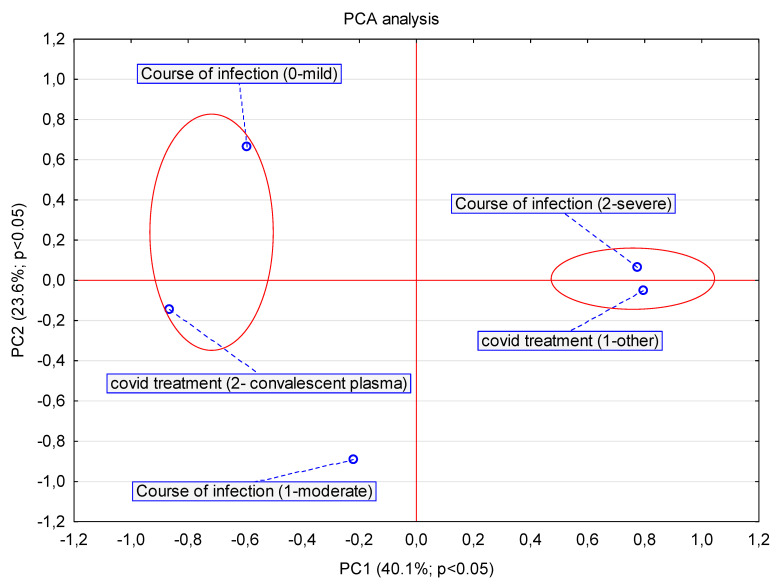
The PCA analysis presented in the pca1 vs. pca2 load diagram. Analysis of the effect of applied early plasma therapy on the course of COVID-19 in hematological patients using generalized principal component analysis PCA. The PCA model was estimated using the NIPALS iterative algorithm, the convergence criterion was set at the level of 0.00001, setting the maximum number of iterations at 50. The number of components was determined by determining the maximum predictive capability Q^2 using the V-fold cross-validation method, setting the maximum number of components at the level of V_max = 7. The obtained optimal PCA model was finally reduced to 2 components.

**Table 1 viruses-13-00436-t001:** Baseline clinical characteristics, laboratory data and outcome of patients with COVID-19.

Variable	Treatment Group of Patients, *n* = 23	Control (Historical) Group of Patients, *n* = 22	All Patients, *n* = 45	*p*
**Age Median (Range)**	57 (31–72)	62.5 (20–80)	59 (20–80)	*p* = 0.58647 (OR = 1.422)
**Male *n* (%)**	14 (61)	14 (64)	28 (62)	*p* = 0.84824 (OR = 1.125)
**Female *n* (%)**	9 (39)	8 (36)	17 (38)	
**Diagnosis**				
**Acute Leukemia/MDS EB2 *n* (%)**	14 (61)	9 (41)	23 (51)	*p* = 0.53659
**Chronic Lymphocytic Leukemia/Indolent Lymphoma *n* (%)**	2 (8.7)	4 (18)	6 (13)	
**Aggressive Lymphoma *n* (%)**	4 (17)	4 (18)	8 (18)	
**Multiple Myeloma *n* (%)**	2 (8.7)	4 (18)	6 (13)	
**Other * *n* (%)**	1 (4.3)	1 (4.5)	2 (4.4)	
**Hematologic Malignancy Status**				
**First Line Treatment**	11 (48)	8 (36)	19 (42)	*p* > 0.05
**Relapsed or Progression**	7 (30)	9 (41)	16 (36)	
**Remission**	5 (22)	5 (23)	10 (22)	
**Comorbidities**				
**0 (%)**	4 (17)	3 (14)	7 (16)	*p* = 0.72828 (OR = 1.333)
**1–2 (%)**	5 (22)	3 (14)	8 (18)	*p* = 0.47729 (OR = 1.759)
**≥3 (%)**	14 (61)	16 (73)	30 (67)	*p* = 0.39896 (OR = 1.714)
**Symptoms**				
**Fever *n* (%)**	15 (65)	15 (68)	30 (67)	*p* = 0.00665 (OR = 3.273)
**Dyspnea *n* (%)**	12 (52)	15 (68)	27 (60)	*p* = 0.03008 (OR = 2.032)
**Cough *n* (%)**	13 (57)	10 (45)	23 (51)	*p* = 0.00763 (OR = 3.333)
**Other ** *n* (%)**	6 (26)	12 (55)	18 (40)	*p* = 0.03723 (OR = 2.252)
**COVID-19 Pneumonia *n* (%)**	17 (74)	18 (82)	35 (78)	*p* = 0.02480 (OR = 8.772)
**Laboratory Results**				
**WBC [g/L, m(range)]**	3.4 (0.01–36.02)	3.5 (0.02–44.7)	3.5 (0.01–44.7)	*p* = 0.04738 (OR = 3.02)
**Lymphocytes [g/L, m(range)]**	0.6 (0.01–5.87)	0.6 (0.02–3.5)	0.6 (0.01–5.87)	*p* = 0.10452 (OR = 1.105)
**Neutrophiles [g/L, m(range)]**	2.3 (0.01–11.78)	2.3 (0.01–15.8)	2.3 (0.01–15.8)	*p* = 0.08154 (OR = 2.631)
**Platelets [g/L, m(range)]**	73 (1–473)	79 (1–511)	79 (1–511)	*p* > 0.05
**Hgb [g/dL, m(range)]**	9.65 (4.9–13.2)	9.7 (6.9–13.7)	9.7 (4.9–13.7)	*p* = 0.14940 (OR = 1.412)
**CRP [mg/L, m(range)]**	32 (0.5–306)	32 (2–350)	32 (0.5–350)	*p* = 0.01415 (OR = 1.730)
**SpO_2_ [m(range)]**	97 (70–100)	97 (75–100)	97 (70–100)	*p* > 0.05
**COVID-19 Severity**				
**Mild *n* (%)**	12 (52.2)	6 (27.3)	18 (40)	*p* = 0.03807
**Moderate *n* (%)**	6 (26.1)	3 (13.6)	9 (20)	
**Severe *n* (%)**	5 (21.7)	13 (59.1)	18 (40)	
**Duration of SARS CoV-2 Infection [days, m(range)]**	18 (8–28)	37 (20–53)	21 (8–53)	*p* = 0.00001 (OR = 6.056)
**Treatment**				
**Oxygen Therapy *n* (%)**	15 (65.2)	16 (72.7)	31 (68.9)	*p* = 0.02355 (OR = 1.403)
**High-Flow Nasal Oxygen *n* (%)**	1 (4.3)	5 (22.7)	4 (8.9)	*p* = 0.06983 (OR = 6.471)
**Mechanical Ventilation *n* (%)**	3 (13)	4 (18.2)	7 (15.5)	*p* = 0.39295 (OR = 2.857)
**Fresh Frozen Plasma *n* (%)**	23 (100)	0 (0)	23 (51.1)	N.D.
**Hydroxychloroquine *n* (%)**	0 (0)	22 (100)	22 (48.9)	N.D.
**Dexamethasone *n* (%)**	8 (34.8)	12 (54.5)	20 (44.4)	*p* = 0.18231 (OR = 2.250)
**Other Treatment (Remdesivir, Tocilizumab, Lopinavir/Ritonavir) *n* (%)**	0 (0)	3 (13.6)	3 (6.7)	N.D.
**Clinical Outcome, Death *n* (%)**	3 (13)	9 (41)	12 (27)	*p* = 0.03460 (OR = 4.615)

* ITP—Immune thrombocytopenic purpura, aplastic anemia; ** diarrhea, nausea and vomiting, loss of smell and taste, conjunctivitis; *n*—number; N.D.—Not Determined.

## Data Availability

The data presented in this study are available on request from the corresponding author.

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
