# Peer review of "Early Administration of Convalescent Plasma Improves Survival in Patients with Hematological Malignancies and COVID-19"

_viruses, 2021, doi:10.3390/v13030436_

Round 1

Reviewer 1 Report

Congratulation for your work and your efforts to support with your data advanced therapeutic options for Covid-19 patients with haematological malignancies. we should always encourage investigators who orchestrate efforts and ideas to provide the scientific community with new and significant information. I personally acknowledge that you have contributed laboriously to define your objectives but I have serious worries about the methodology followed and the interpretation of your results. Noteworthy the extensive analysis of your data provided valuable information but however there are some points that I would like to comment and your response will be definitely appreciated.

Comment 1

In the Abstract (line 40) is stated that "In conclusion, we have demonstrated that convalescent plasma is a safe.... ". However in your study it is not obvious anywhere in the results that calculations for plasma administration was a safe procedure because the safety issues deal with the monitoring of the presence for example of possible adverse reactions normalized for several issues including dosages. Investigations for safe administration cannot simply be considered the results of a better disease course. Thus I feel that your text has to be rephrased.

Comment 2

In general the sample size is small and there is no information about the power of the study with the sample size calculations. The historical selection of the control group (line 69) for collecting data could be an origin of bias for comparison. There is also a wide age distribution of the participants from 20 to 80 years old that makes the study populations very heterogeneous. Even more I feel that the pathophysiological causes of the study volunteers varies significantly. Older patients and those in a very acute phase of AML or with a very long history of CLL have diminished immune responses. We not have any information whether in the statistical evaluation the results were normalized to these variables. We are also lacking information about the anti-malignant treatment course, such as of how far or close the volunteers were from chemotherapy and about how many patients received one or multiple plasma infusions (line 72 and 156). We would then like to know if the results were also normalized to these parameters.

In any case I would expect all above to be commented in the discussion section or to be included in the study limitations.

Author Response

We are sincerely grateful to the reviewers for their effort and time invested in evaluating our manuscript. Their insightful comments and suggestions helped us greatly improve our manuscript. All major changes in the revised manuscript are marked in yellow.

Comment 1

In the Abstract (line 40) is stated that "In conclusion, we have demonstrated that convalescent plasma is a safe.... ". However in your study it is not obvious anywhere in the results that calculations for plasma administration was a safe procedure because the safety issues deal with the monitoring of the presence for example of possible adverse reactions normalized for several issues including dosages. Investigations for safe administration cannot simply be considered the results of a better disease course. Thus I feel that your text has to be rephrased.

Response to Comment 1

Thank you for that comment. We agree with you that the safety of administration of convalescent plasma is very important and its administration should be conducted according to similar rules as the administration of other blood products, moreover, it is an experimental method and therefore we have obtained bioethical approval for its administration. Convalescent plasma transfusion may be associated with adverse post-transfusion reactions, including serious ones, i.e.: non-haemolytic post-transfusion reaction (mainly chills, fever), allergic and anaphylactic reactions, including to compounds used or produced in the inactivation process, reaction to citrate if a large volume of plasma is transfused rapidly. Rapid and large volume transfusion may be associated with the complication of circulatory overload.  The risk of post-transfusion acute respiratory failure (TRALI) was reduced due to the qualification for plasma collection of individuals without transfusion or history of pregnancy.   

Therefore, our patients were carefully monitored for 24 hours to assess early post-transfusion reactions as well as late reactions throughout their hospitalization. In our study all patients tolerated convalescent plasma transfusion very well. No adverse post-transfusion reactions were observed in any patient.

In Abstract section the following statement was changed:

“In conclusion, we have demonstrated that convalescent plasma is an effective treatment and its early administration leads to clinical improvement,”

The following statement was added in the Methods section:

“Convalescent plasma transfusion was administered at approximately 10 mL for the first 10 minutes, which was then increased to approximately 200 mL per 30 minutes. The patients were carefully monitored for 24 hours to assess early post-transfusion reactions as well as late reactions throughout their hospitalization.”

Comment 2

In general the sample size is small and there is no information about the power of the study with the sample size calculations. The historical selection of the control group (line 69) for collecting data could be an origin of bias for comparison. There is also a wide age distribution of the participants from 20 to 80 years old that makes the study populations very heterogeneous. Even more I feel that the pathophysiological causes of the study volunteers varies significantly. Older patients and those in a very acute phase of AML or with a very long history of CLL have diminished immune responses. We not have any information whether in the statistical evaluation the results were normalized to these variables. We are also lacking information about the anti-malignant treatment course, such as of how far or close the volunteers were from chemotherapy and about how many patients received one or multiple plasma infusions (line 72 and 156). We would then like to know if the results were also normalized to these parameters.

In any case I would expect all above to be commented in the discussion section or to be included in the study limitations.

Response to Comment 2

Tha you for that comment. We had intended to analyze how the administration of convalescent plasma would affect the already high risk of severe COVID-19 course in our patients. Since there is still no effective treatment for COVID-19, the administration of plasma from recovered patients may help protect the patients with hematologic malignancies from severe complications and death, as it was shown in our study.

Our study was not a designed, randomized study but an observational, clinical and prospective study resulting from the observation of patients admitted to a certain hospital and the authors had no influence on the selection and size of the groups. However, both groups consisted of patients with similar malignancies and similar age, the groups were also similar in type of chemotherapy regimens used.

The results of our study should be considered with caution but they are based on our clinical observations and reliable statistical methods. A nonparametric survival analysis was based on the nonparametric Cox proportional hazards model and the PCA model was estimated using the NIPALS iterative algorithm, the convergence criterion was set at the level of 0.00001. The results have been normalized to all analyzed parameters, including age, disease status, number of chemotherapy cycles.

  1. The description of the statistical method has been corrected in Methods Section:

 The PCA model was estimated using the NIPALS iterative algorithm, the convergence criterion was set at the level of 0.00001. Guided by the results of PCA principal component analysis, the primary grouping variable (Plasma YES/NO) was determined. Univariate logistic regression with a free expression ≠ 0 was used to determine its correlation with the other key variables of the study. The parameters of the logit function were estimated by the least squares method using the Quasi-Newton algorithm. The criterion of convergence for the performed logistic regression was set at the level of 0.00001. Odds ratios OR were calculated for each comparison along with their 95% significance interval. The statistical significance of the logistic models was evaluated using the chi-square test for degrees of freedom df=1 and an assumed significance level α=0.05.

  1. The following sentences were added to the manuscript in Results and Discussion Section:

-In the treatment group, twenty-one patients received one plasma transfusion and supportive care, whereas two patients received two doses of convalescent plasma.

-This study has several limitations. First, this study included relatively small group of patients with subgroups treated in first and second wave of pandemic. International multicenter randomized trails in patients with hematologic malignancies are needed. Since there is still no effective treatment for COVID-19, the administration of plasma from recovered patients may help protect patients with hematological malignancies from severe complications and death, as shown in our study. The second, our study concerns a heterogeneous patients population with various different haematologic malignancies at different stages of anticancer treatment, further studies are needed to optimize the best time and dosage of convalescent plasma in combination with other antiviral drugs in patients with specific malignancies.

- The hematologic malignancy status has been added to Table 1.

Reviewer 2 Report

The manuscript by Biernat et al. presented the effectiveness con convalescent plasma in patients with hematological malignancies and COVID-19. The paper is well organized with a detailed description of participants and controls, however there is a major concern with the paper.

As authors described, they included a control group (n=22) that was treated during the first wave of COVID-19 (March-May 2020) while the convalescent plasma group (n=23) was treated during the second wave of pandemic (September- November 2020).

  1. The prognosis of COVID-19 patients has changed over the course of the pandemic including more clinical experience, different management and outcomes. The authors should discuss the inclusion of the control group (“historical” control) and the repercussions in the study.
  2. Limited number of patients included in both groups to support the author’s conclusions. This study should have been a randomized controlled trial.

Author Response

We are sincerely grateful to the reviewers for their effort and time invested in evaluating our manuscript. Their insightful comments and suggestions helped us greatly improve our manuscript. All major changes in the revised manuscript are marked in yellow.

The response to comment 1:

Thank you very much for your comment. We agree that the course of the pandemic has changed, there are different diagnostic and therapeutic approaches and outcomes are also different at the beginning of the epidemic and now, however, we still do not have an effective treatment against this virus, too many patients are dying in the course of COVID-19 and what we wanted to show in our study that haematological patients before and now represent a group at high risk of death, that is why they need a special attention. In these patients, there is a need for rapid diagnosis of the SARS-CoV-2 infection and rapid implementation of treatment, and the early administration of convalescent plasma can make a significant difference to the fate of these patients.

We have added the paragraph in Discussion Section according to your suggestion:

Our study has several limitations. First, the study was conducted in a small cohort of patients, with subgroups treated in first and second wave of pandemic. International multicenter randomized trials in patients with hematologic malignancies are needed. Since there is still no effective treatment for COVID-19, the administration of plasma from recovered patients may help protect patients with hematological malignancies from severe complications and death, as shown in our study. The second, our study concerns a heterogeneous patients population with various different haematologic malignancies at different stages of anticancer treatment, further studies are needed to optimize the best time and dosage of convalescent plasma alone or in combination with other antiviral drugs in patients with specific malignancies.

The response to comment 2

We agree with you that the analyzed group of patients is relatively small but as we mentioned in our study, this is the first report to analyze the efficacy of convalescent plasma in a cohort of patients with hematological malignancies.

Our study was not a designed study but an observational, clinical and prospective study resulting from the observation of patients admitted to a certain hospital and the authors had no influence on the selection and size of the groups.

The results of our study should be considered with caution but they are based on our clinical observations and reliable statistical methods. A randomized control trail has already been implemented and we belive that the results will be available soon.

Reviewer 3 Report

Major revisions: This manuscript is of a high quality and scientific soundness and thus no suggestion for major revisions are being made.

Minor revisions:

  • Typos/missing words in line 45, 82, figure 2 table title
  • Please give the exact name and producer of the SARS-CoV-2 RBD IgG-ELISA.

Author Response

We are sincerely grateful to the reviewers for their effort and time invested in evaluating our manuscript. Their insightful comments and suggestions helped us greatly improve our manuscript. All major changes in the revised manuscript are marked in yellow.

Thank you for your remarks.

  • Typos/missing words in line 45, 82, figure 2 table title (tu nie mogÅ‚am znaleźć tych literówek!)

The manuscript has been checked by the English corrector and errors have been corrected according to your suggestions.

  • Please give the exact name and producer of the SARS-CoV-2 RBD IgG-ELISA.

Thank you for the suggestion.

The following sentence was added: (Elecsys® Anti-SARS-CoV-2, ROCHE).

Round 2

Reviewer 2 Report

Thank you for the work invested to overcome our observations and comments. The authors improved their manuscript, this reviewer has no further suggestions besides the following references to strengthen the discussion section.

Ferrari S, Caprioli C, Weber A, Rambaldi A, Lussana F. Convalescent hyperimmune plasma for chemo-immunotherapy induced immunodeficiency in COVID-19 patients with hematological malignancies. Leuk Lymphoma 2021;0(0):1–9.

Tremblay D, Seah C, Schneider T, et al. Convalescent Plasma for the Treatment of Severe COVID-19 Infection in Cancer Patients. Cancer Med 2020;9(22):8571–8.

Hueso T, Pouderoux C, Péré H, et al. Convalescent plasma therapy for B-cell-depleted patients with protracted COVID-19. Blood 2020;136(20):2290–5.